# Evaluation of Toxicity and Efficacy of Inotodiol as an Anti-Inflammatory Agent Using Animal Model

**DOI:** 10.3390/molecules27154704

**Published:** 2022-07-23

**Authors:** Thi Minh Nguyet Nguyen, So-Young Ban, Kyu-Been Park, Chang-Kyu Lee, Seoung-Woo Lee, Young-Jin Lee, Su-Min Baek, Jin-Kyu Park, My Tuyen Thi Nguyen, Jaehan Kim, Jihyun Park, Jong-Tae Park

**Affiliations:** 1CARBOEXPERT Inc., Daejeon 34134, Korea; nguyet@carboexpert.com (T.M.N.N.); syban@carboexpert.com (S.-Y.B.); kbpark@carboexpert.com (K.-B.P.); cklee@carboexpert.com (C.-K.L.); 2Department of Food Science and Technology, Chungnam National University, Daejeon 34134, Korea; 3Department of Veterinary Pathology, College of Veterinary Medicine, Kyungpook National University, Daegu 41566, Korea; pyrk2000@gmail.com (S.-W.L.); bnm3448123@naver.com (Y.-J.L.); suminbaek@naver.com (S.-M.B.); jinkyu820@knu.ac.kr (J.-K.P.); 4Department of Food and Nutrition, Chungnam National University, Daejeon 34134, Korea; mytuyen108@gmail.com (M.T.T.N.); jaykim@cnu.ac.kr (J.K.)

**Keywords:** inotodiol, toxicity, inflammation, allergy, mast cell

## Abstract

Chaga mushroom (*Inonotus obliquus*) comprises polyphenolic compounds, triterpenoids, polysaccharides, and sterols. Among the triterpenoid components, inotodiol has been broadly examined because of its various biological activities. The purpose of this study is to examine inotodiol from a safety point of view and to present the potential possibilities of inotodiol for medical usage. From chaga mushroom extract, crude inotodiol (INO20) and pure inotodiol (INO95) were produced. Mice were treated with either INO20 or INO95 once daily using oral administration for repeated dose toxicity evaluation. Serum biochemistry parameters were analyzed, and the level of pro-inflammatory cytokines in the serum was quantified. In parallel, the effect of inotodiol on food allergic symptoms was investigated. Repeated administration of inotodiol did not show any mortality or abnormalities in organs. In food allergy studies, the symptoms of diarrhea were ameliorated by administration with INO95 and INO20. Furthermore, the level of MCPT-1 decreased by treatment with inotodiol. In this study, we demonstrated for the first time that inotodiol does not cause any detrimental effect by showing anti-allergic activities in vivo by inhibiting mast cell function. Our data highlight the potential to use inotodiol as an immune modulator for diseases related to inflammation.

## 1. Introduction

Chaga mushroom (*Inonotus obliquus*) comprises polyphenolic compounds and triterpenoids such as inotodiol [1]. These compounds have been reported to have broad bioactivities, such as cell cycle inhibition, anticancer activity, and anti-inflammatory effects [2,3]. As chaga mushroom gains increasing attention, its biological activities have been researched widely, and it has been uncovered to have antiviral, antioxidant, immunomodulatory, hypolipidemic, and hypoglycemic activities [4,5,6,7,8,9]. For many ages, chaga mushroom has been traditionally used as a folk medicine in several countries for diabetes, tuberculosis, and cardiovascular disorders, and the mechanisms of these biological activities have been studied to support its medicinal purposes [10,11]. For example, it has been investigated that oral administration of chaga mushroom extract inhibits LPS-induced NF-kB activation, and it demonstrates the mechanism of anti-inflammatory activity. In experimental colitis, the extracts exerted anti-inflammatory effects and regulated lipid metabolism [11,12,13]. Moreover, mast cell-induced anaphylactic response could be attenuated by treatment with chaga mushroom extract [14,15]. In a previous study, it was illustrated that chaga mushroom extract has an effect on a variety of immune responses. It influenced cytokine production of Th2 cells, immunoglobulin production (IgG1, IgE, and IgA) by antigen-specific B cells, and the function of mast cells in food-allergic models induced by orally administered cOVA, even though the effects on Th2 cytokine production still remain unclear [16].

In addition, inotodiol (Figure 1A) also modulates the functions of dendritic cells (DCs) in terms of the expression level of maturation surface markers on DCs [17]. Although the beneficial bioactivity of inotodiol has attracted attention in recent years, the toxicological and safety of inotodiol have not been investigated.

The safety of inotodiol is of importance when considering it as a functional food ingredient. The purpose of this study is to examine inotodiol from a safety point of view and to present the potential possibilities of inotodiol for medical application. 

Mice were repeatedly fed with pure inotodiol (95% purity, INO95) or crude (20% inotodiol, INO20) for 13 weeks. Toxicity on the liver and lipid metabolism were intensively evaluated in addition to the general toxicity studies. Furthermore, the efficacy of inotodiol on food allergy was examined with a new formulation for oral administration.

INO95 did not cause any detrimental effect by showing anti-allergic activities in vivo by regulating the functions of mast cells. Here, we demonstrate the potential of INO95 as an immune modulator for diseases related to inflammation. Further studies are necessary to elucidate clear mechanisms of inotodiol for anti-allergic effects for a better understanding of its therapeutic efficacies.

## 2. Results

### 2.1. Repeated dose Toxicity Study

#### 2.1.1. Effects of Inotodiol on Body Weights 

Toxicity studies were performed in mice up to 13 weeks of treatment with INO95 at doses up to 20 mg/kg and INO20 at a dose of 30 mg/kg did not show any mortality or abnormalities in organs (data not shown). Normal increases were shown in body weight changes, and there was no significant difference in body weight compared to the vehicle group on 4, 8, and 13 weeks (Figure 1B). 

#### 2.1.2. Effects of Inotodiol on Hepatopoietic System 

The treatment of INO95 at doses of 5 and 20 mg/kg, and 30 mg/kg for INO20, respectively, for 13 weeks showed no significant differences in hematological parameters such as the percentage of neutrophils, lymphocytes, monocytes, eosinophils, and basophils compared to the vehicle group (Table 1).

#### 2.1.3. Effects of Inotodiol on Biochemical Parameters and Histology

Repeated administration of INO95 and INO20 did not show any abnormalities in terms of biochemical parameters. No elevations in liver enzymes (AST and ALT) and bilirubin values were observed in mice treated with INO95 and INO20, and histological examination did not show any significant damage in the liver compared to the vehicle group (Table 2 and Figure 2). Hepatic fibrosis was additionally evaluated using Ishak’s modified scoring system to investigate if INO95 and INO20 induce hepatotoxicity. As shown in Figure 3, the score was not altered by the administration of INO95 and INO20. 

Metabolic parameters (triglyceride, cholesterol, and glucose) did not change significantly compared with the vehicle group. In order to investigate lipid metabolism further, LDL and HDL levels were examined, but there were no INO95- or INO20-related effects on LDL and HDL levels (Figure 4).

#### 2.1.4. Effect of Inotodiol on Cytokine Production 

Serum samples from mice treated with INO95 (20 mg/kg) or the vehicle for 13 weeks were used to evaluate the impact of INO95 on cytokine production. INO95 did not affect the cytokine production level in IFN-g, IL-1beta, IL-2, TNF-alpha, IL-6, IL-10, IL-12, and IL-17A. All cytokine levels investigated in this study were almost equivalent to that of the vehicle group (Figure 5).

#### 2.1.5. Prediction of Toxicity and ADME of Inotodiol

When using two different web-based tools, inotodiol properties focusing on toxicity were predicted [18]. ADME parameters are listed in Appendix A and rat toxicity (oral). Ames toxicity and hepatotoxicity predictions were presented in Appendix A. Inotodiol was not expected as a substrate for cytochrome P450s listed in Appendix A. In terms of intestinal absorption and distribution, there were contradictory results between pkCSM and swissADME.

No Ames toxicity of inotodiol was predicted, and inotodiol may not inhibit the hERG I and hERG II. Hepatotoxicity of inotodiol was expected in silico, but no toxicity of inotodiol on the liver was found in 13-week repeated dose experiments.

The probability map of inotodiol for cardiac toxicity prediction was obtained using pred-hERG (Appendix A) [19]. Fragments’ positive and negative contributions to the hERG blockage have been found. The intensity of the pink color shows a negative contribution of an atom or fragment to the hERG blockage. Inotodiol was predicted as potential cardiotoxic with a 60% confidence level according to the pred-hERG. However, there has been no experimental report on the cardiac toxicity of oxysterols which have similar structures to inotodiol.

### 2.2. Effects of Inotodiol on Food Allergic Symptoms 

#### 2.2.1. Overall Symptoms

The oral challenge of cOVA resulted in a drop in body temperature in the sham group, but mice treated with inotodiol showed no alterations in body temperature compared to the untreated control group (Figure 6D). Similarly, the sham group developed diarrhea in response to the oral challenge of cOVA, but the symptoms of diarrhea were ameliorated by administration with INO95 in a dose-dependent manner. Regarding INO20 treated group (G6), the occurrence of diarrhea started to decrease after treatment with INO20 at the same level as G5 (Figure 6C).

#### 2.2.2. Effects of Inotodiol on Mast Cells

Mast cells play roles in allergic response by releasing various molecules such as MCPT-1 upon activation by allergens, and the number of mast cells changes if immune systems start to respond to allergens. As expected, high numbers of mast cells were marked in both intestine and stomach of the sham group, with an average of 19 mast cells/HPF in the intestine and 26 mast cells/HPF in the stomach, respectively. The numbers of those cells were significantly decreased when mice were treated with either INO95 at the dose of 6 mg/kg (11 cells/HPF in intestine, 19 cells/HPF in stomach), 10 mg/kg (7 cells/HPF in intestine, 10 cells/HPF in stomach). Similar to this, treatment with INO20 at a dose of 10 mg/kg resulted in a significant reduction in the average number of mast cells in the intestine and stomach (8 cells/HPF and 14 cells/HPF) compared to the sham group (Figure 7A).

## 3. Discussion

Inotodiol is known to have various biological activities, including anticancer effects, anti-diabetes, immune-modulatory properties, and anti-inflammatory activities [20]. The prevalence of food allergy has been increasing in many countries, and it imposes a huge burden on the quality of life of patients with symptoms of food allergies [21]. With the rising prevalence of food allergies, effective management is highly required. 

Recently, anti-allergic effects of inotodiol were investigated, and it was demonstrated that by treating with inotodiol, recurrence of symptoms related to allergy was prevented in animal models [15]. These results imply that inotodiol might be beneficial as a medication or a supplement for patients with allergies. Despite the advances in the clinical effects of inotodiol, studies on the safety of inotodiol as a potential therapeutic are limited. Traditional medicines have the potential to improve allergic symptoms but can trigger a broad range of adverse events at the same time [22]. Thus, it is worthwhile to investigate the safety and toxicology of inotodiol. In this study, we focused mainly on toxicological evaluations of inotodiol to testify to its practical usage.

Corticosteroids are one of the widely used treatments for several inflammatory disorders [23,24,25]. In spite of their beneficial effects, side effects associated with long-term treatment with corticosteroids have been reported, and a new approach for the treatment of inflammatory diseases with reduced side effects is highly needed. Previous studies reported that a 14-day repeated dose of dexamethasone significantly decreased the number of lymphocytes in the mesenteric lymph node and spleen as well as the body weight of mice [16,26].

With the increasing use of herbal medicines, several side effects of herbal products have been reported, and liver injury is one of the common side effects which can potentially lead to death [27]. In terms of the impact of inotodiol on liver function, the serum enzyme level and histological examinations did not show any changes compared to the vehicle group, and these data indicate that repeated administration of inotodiol does not damage the physiology and cellular structures of the liver. Since long-term exposure to inotodiol might cause liver dysfunctions, further examination was carried out with the liver using the Ishak scoring system, and the results remained close to the vehicle group. 

There was no impact on lipid metabolism by treatment with inotodiol in this study. Regarding lipid metabolism, it has been reported that oxysterol, one of the LXR agonists, activates liver X receptors (LXRs) and regulates cellular lipid metabolism, which results in increased fatty acid and triglyceride (TG) synthesis [28,29]. In addition, reducing effects of chaga mushroom on total and low-density lipoprotein-cholesterol are shown in a diabetic animal model [30]. It is assumable that in normal condition, no notable changes in lipid metabolism is caused by inotodiol, which is different from what happened in diabetic models by administration of chaga mushroom. Furthermore, by purifying and extracting other components from chaga mushrooms, inotodiol might show different effects regarding lipid metabolism from whole chaga mushrooms. However, it should be further investigated if administration of inotodiol activates LXRs and impacts LXR/RXR target genes, even though no changes were shown in lipid metabolism in this study. 

Also, no hematological and other biochemical parameter alterations were observed in inotodiol-treated mice groups. These repeated dose study data imply that administration of INO95 up to 20 mg/kg and INO20 up to 30 mg/kg is not toxic to mice. 

According to the in silico ADME study, inotodiol was predicted to cause hepatotoxicity, but our liver toxicity data indicated that the inotodiol tested here was non-toxic. In addition, inotodiol showed no or less BBB permeability which indicates inotodiol might not induce adverse effects on the central nervous system (CNS). In terms of metabolism, inotodiol predicted that it did not inhibit any major cytochrome (CYP) enzymes, and it is assumable that drug–drug interactions might not occur because of inotodiol. However, further investigation should be implemented to elucidate the toxicity of inotodiol with various doses.

In anti-allergic activity, the improvement of diarrhea occurrence in mice treated with 10 mg/kg of INO20 was similar when treated with 10 mg/kg of INO95. Given the fact that INO20 is a mixture that includes only 20% of inotodiol, other components included in INO20 other than inotodiol might exert biological activities. Betulin, which is included in INO20 (Appendix A), shows diverse biological activities [31], and it might be one of the components that exerted anti-inflammatory activity together with inotodiol when INO20 was administered [32]. Further study is necessary to investigate other triterpenoid components (Appendix A) that impact anti-inflammatory activities in INO20. 

The number of mast cells in both the intestine and stomach is reduced by treatment with inotodiol, and it is consistent with a previous study [16]. However, it is still not clear why the number of mast cells was reduced by treatment with INO95. The mechanisms underlying the anti-allergic effects of INO95 remain largely unclear. In terms of the impact on mast cells of INO95, it is possible that INO95 may influence the expression of integrins and chemokine receptors and affects the ability of migration of mast cells in tissues [33]. In conjunction with the down-regulation of MCPT-1 by INO95, changes in these molecules might result in immune modulations in allergen-stimulated mice and consequently lead to improvement of allergic symptoms. It is also conceivable that INO95 suppresses the proliferation of mast cells. Further studies are necessary to elucidate the mechanism of INO95 to affect the functions of mast cells. 

## 4. Materials and Methods

### 4.1. Preparation of Inotodiol

Chaga mushroom powder was purchased from Jungwoodang Company (Seoul, South Korea). Thirty kilograms of the mushroom powder was submerged in food-grade ethanol (300 L) (Daehan Ethanik Fife Co., Ltd., Gyeonggi-do, South Korea) for 20 h at 50 °C. After extraction, the solution was collected and filtered twice (1st: nonwoven fabric, 2nd: membrane filter, 1 µm, 250 mm) and concentrated 10 times under a rotary vacuum to yield a dark brown slurry. For crude inotodiol preparation, the chaga mushroom extract solution was mixed with distill water (DW) in a 1:1 ratio and incubated in a refrigerator at 4 °C for 1 h. The mixture was centrifuged to remove the supernatant and followed by washing four times with twice the initial amount of DW to dissolve precipitated. The final precipitate was collected by centrifugation and dried completely through a freeze dryer (MCFD8508, Ilshin Biobase, Gyeonggi-do, South Korea). The inotodiol content in final powder was confirmed by HPLC (Model 1525, Waters, USA)-ELSD (SofTA 200, Teledyne ISCO, Lincohn, NE, USA), which was 20% purification (INO20).

High purity of inotodiol was purified using HPLC as described previously [15] with modifications. Briefly, INO20 was dissolved in methanol (MeOH) and incubated in the ultrasonic water bath (LK-U225D, LK Labkorea, Gyeonggi-do, South Korea) for 20 min. This material was purified by HPLC system (LC-Forte/R, YMC, Kyoto, Japan) on a glass column (50 × 500 cm) packed with ODS-AQ C18 (20 μm, YMC, Kyoto, Japan) with distilled water (solvent A) and methanol (solvent B) as the mobile phase. The flow rate was 30 mL/min and the gradient method: (0–30 min; 80% B), (30–40 min; 80–90% B), (40–50 min; 90% B), (50–60 min; 90–100% B), (60–150 min; 100% B). Fractions were isolated on a gel CC eluting and identified inotodiol purity using an Ultimate 3000 UHPLC system (Thermo Fisher Scienctific, Idstein, Germany). Inotodiol was dried to powder using a freeze dryer and reconfirmed by HPLC-ELSD, which was 95% purification (INO95).

### 4.2. Animals

Male BALB/C mice, male ICR mice were purchased from SAMTACO Bio Korea (Osan, Korea). Mice were maintained under a specific pathogen-free condition. All animal experiments were performed in compliance with a protocol approved by the Institutional Animal Care and Use Committee of Chungnam National University (CNU). 

### 4.3. Repeated dose Toxicity Evaluation

For in vivo experiment, INO95 and INO20 were completely dissolved in corn oil. The 5-week male ICR mice were divided into 4 groups. In treatment groups, mice were treated with either INO95 or INO20 once daily for 13 weeks (from day 1 to 90) (Figure 8). Vehicle group (G1) was administered with corn oil. Behavior activities and death were observed once daily after administration of compounds during the testing period. Body weight was recorded every 4-week and at the end of the study. Blood samples were collected from the mice every 4-week and serum was prepared for cholesterol test using HDL and LDL/VLDL cholesterol assay kit (Abcam, Cambridge, UK). 

On day 91, mice were sacrificed, and blood samples were collected in anticoagulant herapin tubes and centrifuged to obtain serum for hematological parameters, clinical biochemistry analysis, and cytokine measurement. Liver samples were harvested for histological analysis. 

#### 4.3.1. Hematology and Clinical Biochemistry 

In order to determine hematological parameters, a colter Diff Hematology Analyzer (Beckman Coulter Corporaion, Ramsey, MN, USA) was applied to measure the following parameters: neutrophils, lymphocytes, monocytes, eosinophils, and basophils differential counts. 

Serum biochemistry parameters, including alanine aminotransferase (ALT), aspartate aminotransferase (AST), alkaline phosphatase (ALP), total protein (TP), albumin (ALB), glucose (GLU), blood urea nitrogen (BUN), creatinine (SCr), total cholesterol (CHO), triglyceride (TG), total bilirutin, and calcium, were analyzed using a Hitachi 7080 automatic clinical analyzer (hitachi High-technologies corporation, Tokyo, Japan). 

#### 4.3.2. Measuring Blood Cytokines Using Multiplex Immunoassay 

The multiplex cytokine assay was carried out directly in a 96-well black plate (Bio-Plex Pro, Hercules, CA, USA) at room temperature. The level of pro-inflammatory cytokines in the serum (IFN-γ, IL-1β, IL-2, TNF-α, IL-6, IL-10, IL-12, and IL-17A) were quantified using Bio-plex procytokine, chemokine and growth factor assay kit (Bio-Rad, Hercules, CA, USA) in accordance with manufacture instructions. 

### 4.4. Pharmacokinetic Prediction

#### 4.4.1. Cardiotoxicity Prediction 

In order to predict cardiac toxicity, a free accessible online service pred-hERG 4.2 (http://predherg.labmol.com.br (accessed on 14 June 2022)) was used for its early detection of potential hERG blockers and non-blockers.

#### 4.4.2. Pharmacokinetic of Prediction

For the prediction of inotodiol on ADMET (absorption, distribution, metabolism, exertion, and toxicity), swiss ADME (http://www.swissadme.ch/index.php (accessed on 14 June 2022)) [18], and pkCSM (http://biosig.unimelb.edu.au/pkcsm/prediction (accessed on 14 June 2022)) were used.

### 4.5. Experimental Mouse Model of Food Allergy

Food allergy was sensitized in 5-week-old male BALB/C mice with chicken Ovalbumin (cOVA) as a model allergen as described previously [15,34,35]. Different doses of INO95 or INO20 were orally administered. Diarrhea was assessed by visually monitoring mice up to 1 h following the fifth cOVA challenge. The change in rectal temperature was measured 1 h after the fifth cOVA challenge. On day 42, mice were sacrificed and blood, small intestine, and stomach samples were harvested for ELISA and histology experiments. 

#### 4.5.1. Preparation of the Tissue (Small Intestine) Extract

The small intestine was collected one day after the fifth cOVA challenge and stored immediately in −80 °C. The tissue sample was ground by liquid nitrogen (N_2_), followed by mixing with 500 µL of the extraction buffer (10 mM of Tris-HCl pH 8.0, 150 mM of NaCl, 1% Triton-X-100, 0.5% sodium deoxylcholate, 1 mM of EDTA, 10% glycerol, 1 mM of PMSF, and a cocktail of freshly prepared protease inhibitors). After incubation (for 1.5 h at 4 °C), the tissue lysate was centrifuged at 14,000× *g* rpm for 30 min at 4 °C. The supernatant was then aliquoted and saved at −70 °C until to test MCPT-1 level

#### 4.5.2. Measurement of the Levels of MCPT-1

MCPT-1 concentration in tissue extraction was measured by eBioscience ELISA kit (San Diego, CA, USA).

Blood samples taken from the tail vein were centrifuged at 4000× *g* rpm/min for 15 min at 4 °C to obtain serum. Individual serum samples were stored at −80 °C until analysis. Measurement of MCPT-1 levels in serum were performed with a commercially available eBioscience ELISA kit. 

### 4.6. Histology

Histological samples immediately fixed with 10% formalin overnight and embedded in paraffin. Thin sections (5 µm thick) were prepared from the paraffin-embedded tissue samples in a microtome (Leica Biosystem, Nussloch, Germany). With food-allergic experiment, small intestine and stomach were stained by toluidine blue for mast cells. Stained sections were observed under the microscope and numbers of mast cells in high power field (HPF) were quantified as previously described [16]. In repeated dose toxicity study, livers were stained by hematoxylin and eosin (H&E) for microscopic examination. 

The histopathological analysis of liver injury was performed using Ishak grading system (REFER, PMID; 7560864). The Ishak activity grade was defined as the average of the other scores including confluent necrosis, focal lytic necrosis and portal inflammation. Each histopathologic parameter was estimated in the randomly chosen fields at 200× magnification. 

### 4.7. Statistical Analysis

The significance of the difference between the groups was calculated with nonparametric one-way ANOVA (Kruskal–Wallis test), and also confirmed with Bonferroni correction [36] as post hoc analysis. Both *p* < 0.01 and *p* < 0.05 were considered statistically significant and denoted separately. All analyses were performed with GraphPad Prism software (La Jolla, CA, USA).

## 5. Conclusions

In this study, we demonstrated for the first time that INO95 does not cause any detrimental effect with showing anti-allergic activities in vivo by regulating the functions of mast cells. Our data highlight the potential to use INO95 as an immune modulator for diseases related to inflammation. Further studies are necessary to elucidate clear mechanisms of inotodiol for anti-inflammatory effects for better understanding of its therapeutic efficacies.

## Figures and Tables

**Figure 1 molecules-27-04704-f001:**
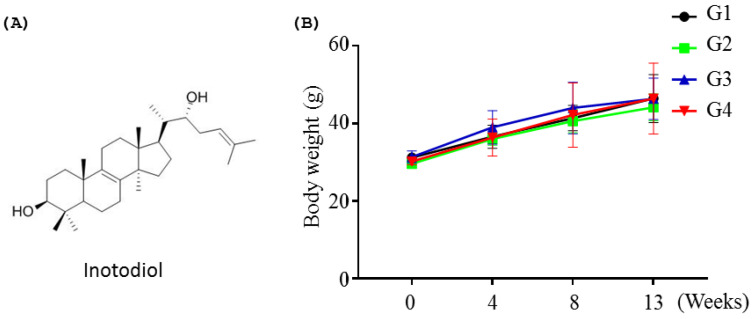
Body weight changes on repeated dose toxicity study. In the experiment, five male ICR mice were administered with or without INO95 and INO20 once a day for 13 weeks. (**A**) The structure of INO95. (**B**) Body weight changes were evaluated for 13-week treatment. The data shown represent mean ± SD (*n* = 5).

**Figure 2 molecules-27-04704-f002:**
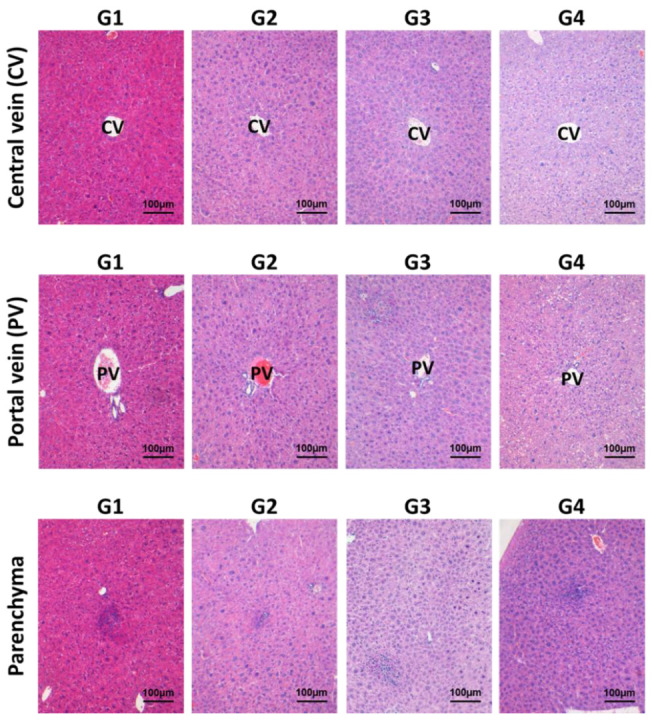
Effects of INO95 and INO20 on hepatitis inflammation in the repeated dose toxicity study. In the experiment, liver was collected on day 91 for histology analysis. Section slides were stained with hematoxylin and eosin (H&E). The inflammatory activities of INO95 and INO20 on liver were examined at 3 different areas such as central vein (CV), portal vein (PV) and parenchyma. Microscopic histology picture for presentation was chosen from the representative pictures of 5 mice (*n* = 5) in a group and viewed at magnifications of ×200.

**Figure 3 molecules-27-04704-f003:**
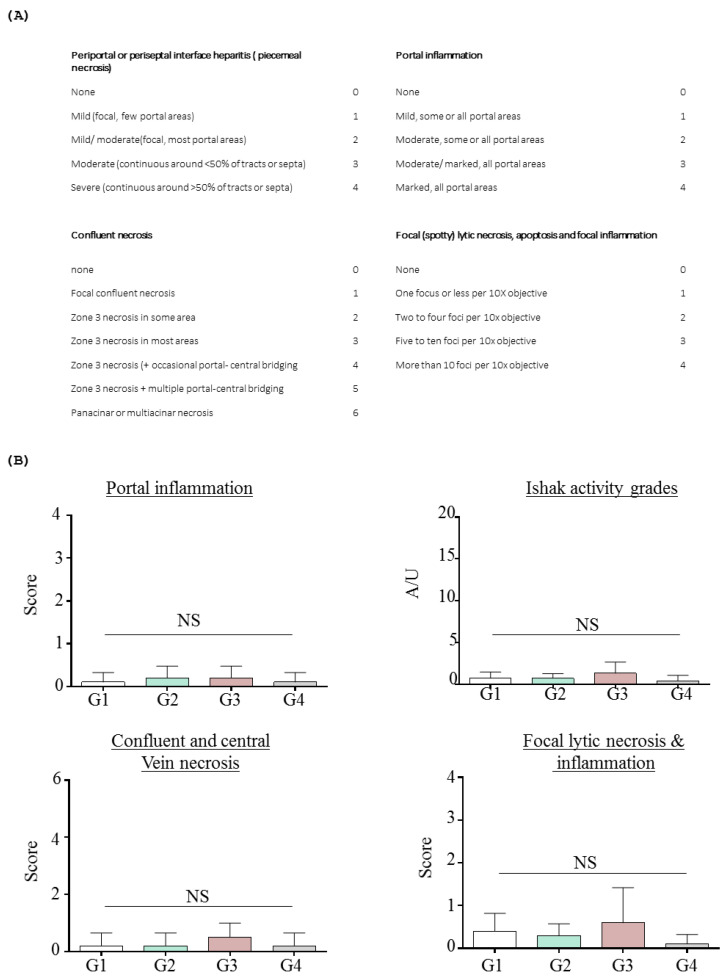
Histological grading of hepatitis inflammation in the repeated dose toxicity study. In the experiment, mice were treated with INO95 and INO20 for 13 weeks. On day 91, livers were collected and analyzed. (**A**) Histology ishak score. (**B**) The grading of histology ishak score between groups. The data shown represent mean ± SD (*n* = 5). Abbreviation: NS: Not significant.

**Figure 4 molecules-27-04704-f004:**
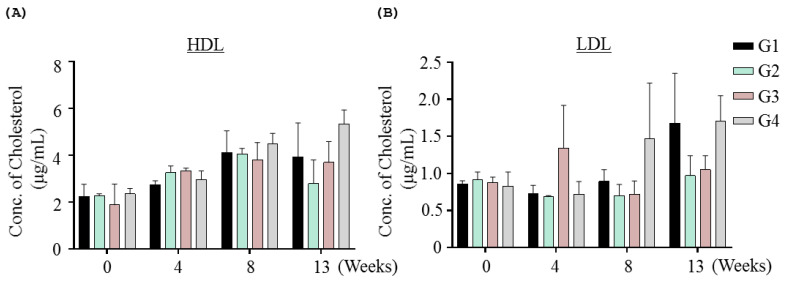
Effect of INO95 and INO20 on cholesterol level in the repeated dose toxicity study. In the experiment, blood samples were collected every four weeks to examine cholesterol level including HDL and LDL. The comparisons of HDL (**A**) and LDL (**B**) level of each group for 13-week treatment are presented. The data shown represent mean ± SD (*n* = 5).

**Figure 5 molecules-27-04704-f005:**
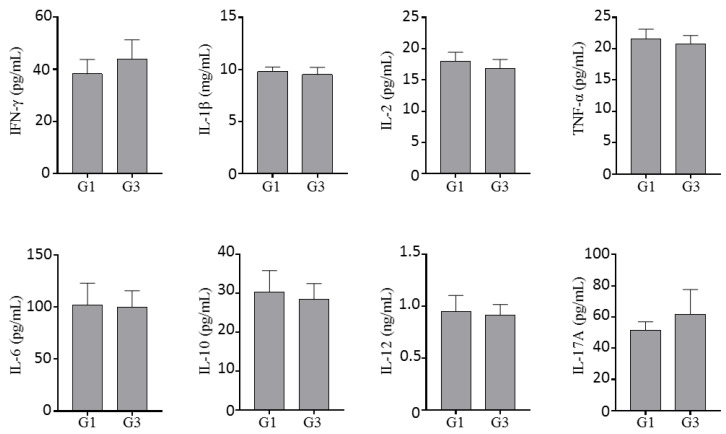
Effect of INO95 on the cytokine production in repeated dosed toxicity study. On day 91, mice were sacrificed, and serum cytokine levels were quantified and compared between INO95 treatment group and the non-treatment group. The data shown represent mean ± SD (*n* = 5).

**Figure 6 molecules-27-04704-f006:**
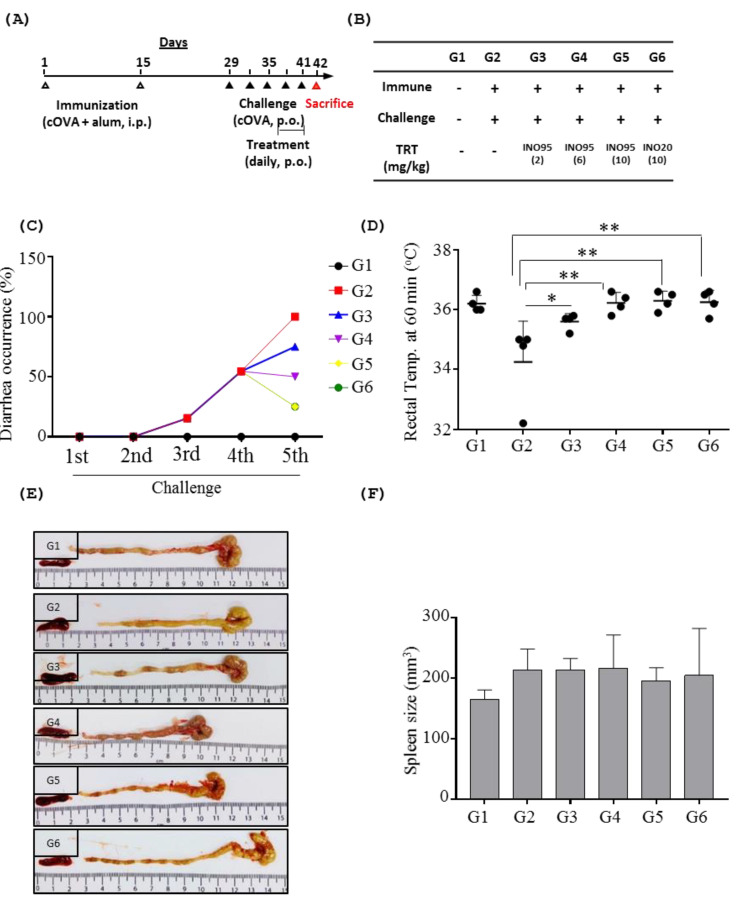
Effects of INO95 and INO20 on food allergic symptoms. Male BALB/C mice were divided into 6 groups and treated with either INO95 or INO20 from day 38 to day 41 once a day. On the day 41, mice were treated with compounds for 1 h, and then were challenged with cOVA. One hour later, clinical symptoms were recorded. (**A**) The experimental scheme is shown. (**B**) The description of each treatment group is shown in the table. (**C**) Diarrhea occurrence was recorded during challenge period. (**D**) Rectal temperature was measured 1 h after the fifth cOVA challenge. On day 42, mice were sacrificed to check organs and blood. (**E**) The morphology pictures of spleen and colon were shown. (**F**) The size of spleen was measured. Significantly different from sham group *p* < 0.05 (*), *p* < 0.005 (**), as determined by one way ANOVA test. The data shown represent mean ± SD (*n* = 4).

**Figure 7 molecules-27-04704-f007:**
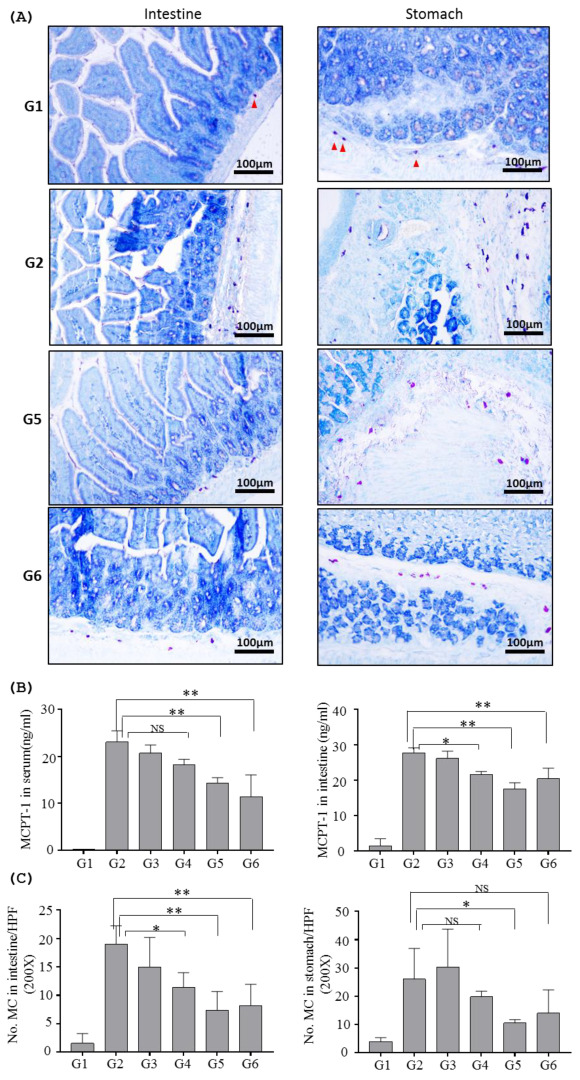
Inhibition by INO95 and INO20 on the function of mast cells in food allergic mouse model. Mice were sacrificed on day 42. Serum was harvested to measure MCPT-1, and small intestine, stomach were collected to analyze cytokine production and histology. (**A**) Mast cells in tissues were determined by toluidine blue staining and counted at magnification of ×200. (**B**) MCPT-1 levels were measured in both serum and tissue extraction. (**C**) Mast cells indicated by red arrow were counted in each group. Significantly different from sham group *p* < 0.05 (*), *p* < 0.005 (**), as determined by one way ANOVA test. The data shown represent mean ± SD (*n* = 4).

**Figure 8 molecules-27-04704-f008:**
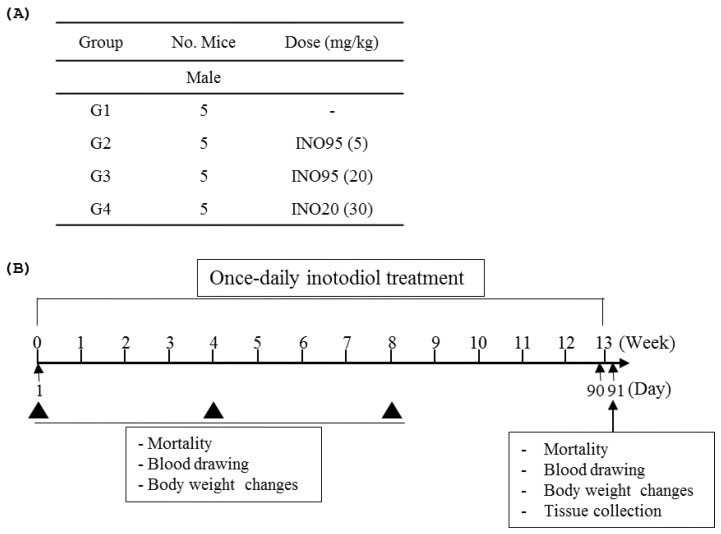
(**A**) Doses of inotodiol and, the number of animals per group of this study are shown. (**B**) Overall experimental design is summarized.

**Table 1 molecules-27-04704-t001:** Hematological result in mice administrated orally with or without INO95 and INO20 of repeated dose toxicity study. The average ± SD is given (*n* = 5).

WBC	Unit	G1	G2	G3	G4
Neutrophils	%	36.9 ± 17.4	31.0 ± 8.8	35.6 ± 11.8	29.3 ± 11.2
Lymphocytes	%	54.1 ± 17.4	60.9 ± 7.4	55.1 ± 11.2	63.9 ± 8.9
Monocytes	%	5.4 ± 2.7	3.9 ± 0.9	5.4 ± 1.9	2.5 ± 1.0
Eosinophils	%	3.2 ± 0.7	4.2 ± 2.2	3.7 ± 3.5	4.3 ± 2.6
Basophils	%	0.38 ± 0.08	0	0.40 ± 0.01	0

**Table 2 molecules-27-04704-t002:** Effects of repeated dose toxicity study of INO95 and INO20 on clinical biochemistry results. The average ± SD of parameters is given (*n* = 5).

	Unit	G1	G2	G3	G4
**Liver function**					
TP	g/DL	6.49 ± 0.6	6.24 ± 0.2	6.21 ± 0.1	6.24 ± 0.40
ALB	g/DL	2.33 ± 0.2	2.18 ± 0.1	2.12 ± 0.1	2.17 ± 0.20
Total bilirubin	mg/DL	0.3 ± 0.06	0.26 ± 0.02	0.25 ± 0.06	0.26 ± 0.03
TG	mg/DL	227.5 ± 52.2	198.2 ± 49.7	220.9 ± 80.7	230.4 ± 102.7
ALP	U/L	162.9 ± 26.7	151.7 ± 35.4	130.1 ± 21.7	130.1 ± 21.7
AST	U/L	112.4 ± 22.2	123.1 ± 39.9	130.0 ± 29.1	108.5 ± 17.2
ALT	U/L	42.5 ± 8.6	33.4 ± 4.1	41.8 ± 6.8	35.3 ± 7.8
CHO	mg/DL	196.3 ± 40.1	206.6 ± 28.7	198.3 ± 23.2	209.9 ± 52.6
GLU	mg/DL	181.3 ± 19.4	201.7 ± 19.5	201.3 ± 29.9	170.0 ± 19.4
**Kidney function**					
SCr	mg/DL	0.49 ± 0.14	0.46 ± 0.04	0.50 ± 0.05	0.48 ± 0.07
BUN	mg/DL	31.1 ± 10.7	31.9 ± 1.1	31.5 ± 4.1	31.0 ± 3.1
Ca	mg/DL	11.5 ± 0.5	11.7 ± 0.31	12.3 ± 0.9	12.3 ± 0.9

## Data Availability

Not applicable.

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
