# Peer review of "Evaluation of Toxicity and Efficacy of Inotodiol as an Anti-Inflammatory Agent Using Animal Model"

_molecules, 2022, doi:10.3390/molecules27154704_

Round 1
Reviewer 1 Report
The subject was well-documented and experiment design was smartly conceived and perfectly performed, so yielding convincing conclusions. But minor revision is required to improve tyhe manuscript, below are few comments and concerns:
- chemical structures of main triterpenoid constituents of chaga mushroom should be drawn as well as their concentrations in crude extract should be given as they may interfere with inotodiol biological activities regarding anti-inflammatory and allergy.
- More comments regarding the expected biological effetcs of other secondary metabolites from chaga mushroom (polypheno compounds, triterpenoids, polysaccharides, sterols) should be presented briefly in introduction part to bring better knowledge about their co-occurence and eventual complementary activities.
So, I recommend the acceptation of this paper for publication in "Molecules " journal once these required revisions done.

Author Response
The manuscript entitled «Evaluation of toxicity and efficacy of inotodiol as an anti-inflammatory agent using animal model» is an orginal study realized to examine Chaga mushroom (Inonotus obliquus) inotodiol component from the safety point of view and to present the potential possibilities of inotodiol for medical usage. Fro that purpose, crude inotodiol (INO20) and pure inotodiol (INO95) were produced from chaga mushroom and orally administrated to treated mice were for repeated dose toxicity evaluation. Serum biochemistry parameters were analyzed and the level of pro-inflammatory cytokines in the serum were quantified. In parallel, the effect of inotodiol on food allergic symptoms was investigated. The subject was well-documented and experiment design was smartly conceived and perfectly performed and so yielding convincing conclusions. But minor revision is required to improve the manuscript. below are a few comments and concerns :
- Chemical structures of main triterpenoids constituents of chaga mushroom should be drawn as well their concentration in crude extract because they may interfere with inotodiol biological activities regarding anti-inflammatory and allergy.
Answer : Thank you so much for your valuable comments. We added the chemical structure of inotodiol which is a main triterpenoid in Figure 1. And data on other compositions of crude inotodiol was added in supplement materials. Since standard molecules are not commercially available, only qualitative analysis was conducted.
- More comments regarding the excpected biological effects of other metabolites from chaga mushroom (polyphenolic compounds, triterpenoids, polysaccharides, sterols) should be presented briefly in introduction part to bring better knowledge about their co-occurrence and eventual complementary activities.
Answer : Thank you for your suggestion. We added more information on biological effects in introduction.
Reviewer 2 Report
The article "Evaluation of toxicity and efficacy of inotodiol as an anti-in- 2 flammatory agent using animal model" is an interesting piece article for readers. The results merit publication. But there are some outcoming in article that needed to be addressed and some suggestions.
1. I will strongly recommend to add the structure of inotodiol as figure 1, even the introduction section or anywhere so at least there is awareness about the structure of that compound.
2. Figure 2 is incorporated in the wrong place. First is Fig.1 then fig3,4, 5, and then figure is incorporated after 5 which is very confusing. Kindly arrange all the figures numerically.
3. The conclusion is missing.
4. As the authors are talking about safety profiles. I will also recommend adding some more data about in-silico studies through ADMET for this compound and also can add additional studies of Cardiotoxicity of the compound through the online free web toll. The references article is attached here. I believe the additional data will make the article more attractive in every aspect. Int. J. Mol. Sci. 2021, 22(21), 11432;
Author Response
The article "Evaluation of toxicity and efficacy of inotodiol as an anti-in- 2 flammatory agent using animal model" is an interesting piece article for readers. The results merit publication. But there are some outcoming in article that needed to be addressed and some suggestions.
- I will strongly recommend to add the structure of inotodiol as figure 1, even the introduction section or anywhere so at least there is awareness about the structure of that compound.
Answer : Thank you so much for your suggestion. We added structure of inotodiol in Figure 1.
- Figure 2 is incorporated in the wrong place. First is Fig.1 then fig3,4, 5, and then figure is incorporated after 5 which is very confusing. Kindly arrange all the figures numerically.
Answer : My sincere apologies for that. We corrected the number of figures.
- The conclusion is missing.
Answer : Thank you so much for your comments. We added conclusion following your advice.
- As the authors are talking about safety profiles. I will also recommend adding some more data about in-silico studies through ADMET for this compound and also can add additional studies of Cardiotoxicity of the compound through the online free web toll. The references article is attached here. I believe the additional data will make the article more attractive in every aspect. J. Mol. Sci.2021, 22(21), 11432;
Answer : Thank you so much for your valuable suggestion. We added in silico ADME and Cardiotoxicity data using online free web tools.
Round 2
Reviewer 2 Report
Dear authors, you almost addressed all my questions very well except the last point.
You have added ADME and toxicity portion in the supplementary but there are many errors in that which are mentioned below.
1. These ADME materials and methods should be written in the materials section and results should be incorporated in results section or supplementary is also ok but at least discuss it in the paper with the section od ADMET.
2. Toxicity portion is missing in the table.
3. In the table distribution is written in front of cytochrome, while all cytochrome are in the category of metabolism. Kindly check the article carefully.J. Mol. Sci.2021, 22(21), 11432;
4. For all ADME sections no reference has been cited. The authors should properly cite the references otherwise it is unethical.
Author Response
- These ADME materials and methods should be written in the materials section and results should be incorporated in results section or supplementary is also ok but at least discuss it in the paper with the section od ADMET.
Answer : Thank you so much for your valuable suggestions. We added ‘in silico ADME and cardiotoxicity predictions’ in material and methods section. The results were presented in results section and supplementary materials as well and we discussed on these results in discussion section.
- Toxicity portion is missing in the table.
Answer: Thank you so much for your comment. We added toxicity portion in table S2.
- In the table distribution is written in front of cytochrome, while all cytochrome are in the category of metabolism. Kindly check the article carefully.J. Mol. Sci.2021, 22(21), 11432;
Answer: Thank you so much for your advice. We revised the table S1 referring to the paper that you kindly shared with us.
- For all ADME sections no reference has been cited. The authors should properly cite the references otherwise it is unethical.
Answer: Thank you so much for your advice. We added related references following your advice.